# Application of the Fuzzy Approach for Evaluating and Selecting Relevant Objects, Features, and Their Ranges

**DOI:** 10.3390/e25081223

**Published:** 2023-08-17

**Authors:** Wiesław Paja

**Affiliations:** Institute of Computer Science, College of Natural Sciences, University of Rzeszów, Rejtana Str. 16C, 35-959 Rzeszów, Poland; wpaja@ur.edu.pl

**Keywords:** attribute and object selection, fuzzification, discretization

## Abstract

Relevant attribute selection in machine learning is a key aspect aimed at simplifying the problem, reducing its dimensionality, and consequently accelerating computation. This paper proposes new algorithms for selecting relevant features and evaluating and selecting a subset of relevant objects in a dataset. Both algorithms are mainly based on the use of a fuzzy approach. The research presented here yielded preliminary results of a new approach to the problem of selecting relevant attributes and objects and selecting appropriate ranges of their values. Detailed results obtained on the Sonar dataset show the positive effects of this approach. Moreover, the observed results may suggest the effectiveness of the proposed method in terms of identifying a subset of truly relevant attributes from among those identified by traditional feature selection methods.

## 1. Introduction

One of the main challenges and goals of machine learning and data-mining methods is to identify the relationships and connections between the features that describe an object or example and the category or class to which that example belongs. In other words, the relationships between the dependent variables and the independent variables. In most real-world datasets, only a subset of the describing features actually have a specific relationship with the dependent variable. This is the set of so-called relevant features, i.e., those carrying information that allows us to identify the value of the dependent variable (class). The rest of the features are referred to as non-relevant features, i.e., those that do not affect the determination of the value of the dependent variable, and only cause an increase in the dimensionality of the problem and, thus, the time complexity of computational methods.

Historically, the relevance of variables has been defined in many different ways that have not been consistent with each other, as demonstrated by Kohavi and John [1]. This happened because different researchers focused on different concepts that could be associated with the term. They proposed using two degrees of relevance (strong and weak) to cover all these concepts. In their approach, relevance is defined in absolute terms, using an ideal Bayes classifier. Thus, a feature a∈A is defined as strongly relevant when removing *a* from the data always results in a degradation of the prediction accuracy of the ideal Bayes classifier. A feature a∈A is weakly relevant if it is not strongly relevant and there is such a subset of features *S* where the performance of the ideal Bayes classifier on *S* is worse than the performance on S∪{a}. A feature is irrelevant if it is neither strongly nor weakly relevant. Rudnicki and Mnich [2] showed that, in practical applications, more detailed definitions of the weak and strong relevance of features may be more appropriate. They proposed the introduction of two concepts, *k*-weak relevance and *k*-strong relevance, which better reflect the limited ability to exhaustively test all possible combinations of variables. Thus, feature *a* is *k*-weakly relevant if its relevance can be determined by analyzing all nk subsets of *k* features that contain feature *a*. Similarly, variable a is *k*-strongly relevant if it is strongly relevant in all nk subsets of *k* features that contain variable *a*.

Many of the non-relevant features are difficult to identify a priori; they may have little effect on the dependent variable, which is difficult to identify. Relevant features may also be redundant and carry the same information, so to speak. In addition, they may be dependent on each other, i.e., a particular feature may show relevance only in the presence of another feature [2].

For the aforementioned reasons, many different feature selection methods, which are more or less specialized, have been developed [3]:Filter-based methods: These methods evaluate the relevance of features independently of the specific machine learning task [4]. Examples include the analysis of variance (ANOVA), Pearson correlation coefficient, informative chi-square coefficient, Gini measure, etc. These methods are fast and model-independent, but may not take into account relationships between features.Wrapper-based methods: These methods use a specific machine learning model as a black box, assessing the quality of features in the context of a given model [5]. Examples include recursive feature elimination, backward stepwise selection, forward stepwise selection, etc. These methods can take into account dependencies between features but are more computationally expensive.Embedded methods: These methods take into account feature selection in the model learning process [6]. Examples include L1 regularization (Lasso), L2 regularization (ridge), decision trees with feature selection, etc. These methods combine the process of model learning and feature selection, which can be more effective, but limits the possibility of model reuse without feature selection.Methods based on principal component analysis (PCA) [7]: PCA is a dimensionality reduction technique that projects data into new non-covariates so as to maximize variance. By selecting the principal components, the dimensions of the data can be reduced. PCA is not a direct feature selection method, but it can help extract relevant information from the data.Methods based on information metrics: These methods measure the informational relationship between features and the target variable. An example is the information gain factor, the expected amount of information (reduction of entropy), which is used in decision trees. These methods help assess what features will contribute the most information to the model. Other similar methods, like the fast correlation-based filter (FCBF), are also entropy-based measures, which additionally identify redundancy due to pairwise correlations between features [8].

Various hybrid feature selection methods and techniques that adapt to specific problems and data are also used in practice [9]. The choice of an appropriate feature selection method depends on the specifics of the problem, the available data, and the requirements for model efficiency and interpretability.

In recent years, there have also been quite a few approaches to feature selection using the fuzzy set theory. This approach has become the basis for the implementation of several different algorithms incorporating fuzzy approaches. Fuzzy decision trees (FDTs) are sources of information that allow the construction of new indicators to evaluate the relevance of features and, on this basis, create their ranking [10]. They proposed importance analysis for the evaluation of the induced classifier properties according to reliability analysis. Similarly, the fuzzy forest approach [11] evaluates and builds a ranking of features in classification and regression problems; this method is, in turn, an extension of the random forest algorithm. Others [12] proposed a new measure of similarity between two linguistic, intuitionistic fuzzy sets to formally define the correlation between attributes. One can also distinguish the fuzzy backward feature elimination (FBFE) method [13], which is based on combining the independent component analysis method, derived, as it were, from the well-known principal component analysis method, with the fuzzy entropy measure [14], in the process of eliminating irrelevant features. Still, another approach using fuzziness for feature selection is the fuzzy quick reduce algorithm based on the fuzzy-rough set theory [15], a data-mining algorithm for decision-making based on incomplete, inconsistent, imprecise, and vague data. The fuzzy-rough set theory is an extension of the fuzzy conventional set theory that supports approximations in decision-making.

## 2. Materials and Methods

One of the key problems is to reduce the input data in such a way as to select only relevant data for further operations. Three types of data reduction can be distinguished: the selection of relevant attributes/features, the selection of relevant ranges of feature values, and the selection of relevant objects from the data. The selection of relevant attributes is a typical step in the process of analysis and machine learning. However, the other two types, which are equally important, are the purpose of this research. Assessing the relevance of value ranges has been the subject of previous research [16,17,18]. On the basis of such an assessment, algorithms for the selection of both significant attributes and significant objects in the data can be defined.

In a variety of machine learning applications, input data are arranged in a tabular form that is called a decision table DT=(U,A,D), where
$U=\left\{u_{1},u_{2},u_{3},...,u_{m}\right\}$ is the non-empty, finite set of *m* cases;$A=\left\{a_{1},a_{2},a_{3},...,a_{n}\right\}$ is the non-empty, finite set of *n* descriptive (condition) attributes that describe cases;$D=\left\{D\right\}$ is the non-empty, finite set of decision attributes that classify cases from *U* to decision classes.

For each attribute, the set of its values is determined. Thus, the algorithm for the fuzzy selection of a subset of relevant features is presented in a pseudocode (see Algorithm 1—*fuzzy feature selection*). The aforementioned DT decision table in which attributes have continuous values is used as input. Several key stages can be distinguished in the algorithm, as described in detail in Section 2.2, Section 2.3 and Section 2.4. First, the input data are discretized, resulting in a set $A_{DISC}$ of discrete values. Then, using the chosen fuzzification method, we determine the membership function and, based on it, the set $A_{FUZZ}$ of fuzzy linguistic variables, *LVs*. The next important step is the selection of significant linguistic variables from the set of $A_{FUZZ}$ using the selected feature selection method; the result is the set of $A_{FUZZ_{SEL}}$. The obtained selection results are used to convert the attribute set *A* into $A_{BINARY}$ binary form, representing the relevance of individual attribute values in a binary manner, considering the $A_{FUZZ_{SEL}}$ set. The binary set is the basis for assessing the relevance and selection of relevant attributes and relevant objects for the *fuzzy object selection* algorithm (see Algorithm 2). To assess the relevance of a given attribute, it is necessary to determine how many relevant intervals are in the binary set. For this purpose, a *thresholdFFS* is defined (see Equation (Equation 1)), which is equal to the product of the *m* number of objects in the set and the *EPS* (epsilon) parameter, which is the interval span.
(1)thresholdFFS=EPS ∗ m,

*EPS* has values ranging from 0 to 1. Using specific values of the *EPS* parameter, we determine the value of the *thresholdFFS*, e.g., for a value of EPS=0.01, the *thresholdFFS* will be the smallest, so all relevant features (*a*) will remain in the selected set *FS*; if the *EPS* parameter is 0.10, then the *thresholdFFS* is 0.1 × 208 = 20.80 (see the first table in Section 3). *ThresholdFFS* is a parameter that defines, so to speak, the space of important ranges of the feature value. If *ThresholdFFS* is *20.80*, then those features that have more than 20.80 important value ranges are considered relevant. In our case, 22 features meet this condition. Such information is contained in the binary file $A_{BINARY}$. A visualization of a fragment of this set is shown in the third figure in Section 2.4. If the *EPS* parameter is 0.11, then the *thresholdFFS* is 0.11 × 208 = 22.88. The set of features whose value number meets the *thresholdFFS* is smaller and amounts to 19 features. As the value of *EPS* increases, the value of the *thresholdFFS* increases, which causes the number of features (*a*) in the selected set to decrease. As *EPS* increases, the optimal *thresholdFFS* value can be selected, which will select the optimal subset of features with the best classification quality values.

The algorithm for the fuzzy selection of a subset of relevant objects works similarly (Algorithm 2). It works horizontally, so to speak, i.e., we use the number of features *n* to determine the *thresholdFOS* (see Equation (Equation 2)), and in a loop, each of the objects *u* from the set *U* is checked. The result of the operation is a subset of relevant objects *OS*.
(2)thresholdFOS=EPS ∗ n,

For example, if the *EPS* parameter is 0.06, then the *thresholdFOS* is 0.06 × 60 = 3.60 (the second table in Section 3). The *thresholdFOS* value determines how many values of a given learning object should be in the range marked as relevant. If, for example, *thresholdFOS* is 3.60, then those that have more than 3.60 important value ranges are recognized as relevant objects. In our case, 109 objects meet this condition. Such information is also in the binary file $A_{BINARY}$. A visualization of a fragment of this set is shown in the third figure in Section 2.4. If the *EPS* parameter is 0.07, then the *thresholdFOS* is 0.07 × 60 = 4.20. The set of objects whose value number meets the *thresholdFOS* is smaller and amounts to 72 features.   
**Algorithm 1:** Fuzzy feature selection.
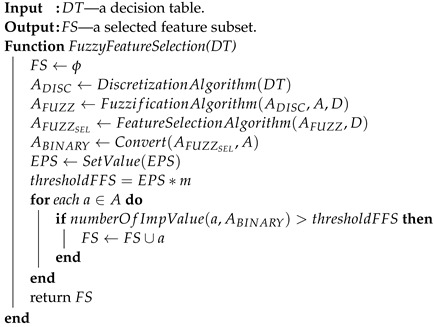


**Algorithm 2:** Fuzzy Object Selection

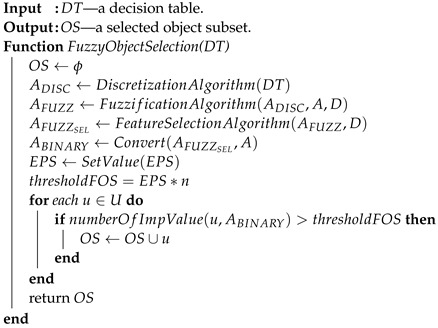



### 2.1. The Dataset Used

The classification quality of the presented feature and object selection algorithms was obtained on a dataset called *Sonar* [19]. This dataset came from the UCI repository and was added to the core collection by Terry Sejnowski (Salk Institute and University of California, San Diego, CA, USA). The tested dataset contains 111 patterns obtained by reflecting off a metal cylinder at different angles and under different conditions. The transmitted sonar signal is frequency-modulated, with increasing frequency. The data include signals obtained at different angles, 90 degrees for the cylinder and 180 degrees for the rock. Each pattern in this dataset consists of a set of 60 numbers, ranging (*V*1–*V*60) from 0.0 to 1.0. Each number expresses the energy in a specific frequency band, which is consolidated over a specific time. The consolidation aperture for higher frequencies takes place later, as these frequencies are sent later via the chirp. In addition, for each record, the data show whether the object is a rock (*R*) or a mine (*M*), i.e., a metal cylinder. The provided operations will be illustrated on the basis of the variable *V9* from the Sonar set and the results of its processing.

Other experiments were conducted to compare the effectiveness of the fuzzy feature selection with other methods on the Pima Indians diabetes database (Pima), the breast cancer Wisconsin diagnostic (BCWD), climate model simulation crashes (Climate), and single proton emission computed tomography (SPECTF) datasets from the UCI repository, which is well-known in the field of machine learning.

### 2.2. Discretization Algorithm

To determine the ranges of descriptive attribute values underlying the fuzzification process (*DiscretizationAlgorithm*), a supervised discretization that depends on local distinguishability heuristics [20] was used. This discretization gives us locally semi-optimal cut sets that are consistent with the input decision table. The cuts divide entire ranges of descriptive attribute values into disjoint sub-ranges that correspond to the linguistic values assigned to these attributes. (see Table 1). In the fuzzification process, the centers of the subintervals are determined and then the membership functions are defined. The local strategy is implemented through a decision tree. This strategy is based on finding the best cut and dividing the set of cases into two subsets of cases, repeating this processing for each set of cases separately until it is satisfied. The quality of the cut depends on the number of cases recognized by the cut, in the local strategy calculated locally on the subset of cases.

### 2.3. Fuzzification Algorithm

One of the methods used is fuzzification, so the idea of a fuzzy set should first be mentioned.

**Definition** **1**([21])**.**
*A fuzzy set R in X≠∅ is*
(3)R={(x,R(x))|x∈X}
*where R:X→[0, 1] and R(x) is the recognized grade of membership of the x to the R. The collection of all fuzzy sets in X will be denoted by FS(X).*

To convert variables with continuous values into linguistic variables *LV* (*FuzzificationAlgorithm*), the selected triangular membership function was used. Let [min,max] be the entire range of values for a given attribute, attribute *a*, of the dataset under study. In the fuzzification process, three steps are performed:For each descriptive attribute *a*, for each linguistic value *l* assigned to *a*, determine the center of the interval corresponding to *l* (the means are calculated based on the intervals on which the entire range of attribute values [min,max] is divided).For each descriptive attribute *a*, for each linguistic value *l* assigned to *a*, define a membership function based on the means of the intervals determined earlier.For each descriptive attribute *a*, calculate the values of the fuzzy descriptive attributes corresponding to *a* based on the membership functions defined previously.

Many different membership functions can be used to determine the linguistic values of individual variables. In the present experiments, they are limited to a typical triangular function, which becomes a trapezoidal function at the edges of the interval.

Let $\left\{c_{1},c_{2},...,c_{k}\right\}$ be the set of interval centers defined for the i-th descriptive attribute.

The triangular membership functions are defined according to Equations (Equation 4)–(Equation 6). In fact, the first and last membership functions are trapezoidal:

For j=1
(4)$$\mu_{c_{j}}\left(x\right)=\left\{\begin{array}{cc} 1, & \mathrm{if}\;x\geq a\;\mathrm{and}\;x\leq c_{j},\\ \frac{c_{j+1}-x}{c_{j+1}-c_{j}}, & \mathrm{if}\;x>c_{j}\;\mathrm{and}\;x\leq c_{j+1},\\ 0, & \mathrm{otherwise}. \end{array}\right.$$

For j>1 and j<k
(5)$$\mu_{c_{j}}\left(x\right)=\left\{\begin{array}{cc} \frac{x-c_{j-1}}{c_{j}-c_{j-1}}, & \mathrm{if}\;x\geq c_{j-1}\;\mathrm{and}\;x\leq c_{j},\\ \frac{c_{j+1}-x}{c_{j+1}-c_{j}}, & \mathrm{if}\;x>c_{j}\;\mathrm{and}\;x\leq c_{j+1},\\ 0, & \mathrm{otherwise}. \end{array}\right.$$

For j=k
(6)$$\mu_{c_{j}}\left(x\right)=\left\{\begin{array}{cc} \frac{x-c_{j-1}}{c_{j}-c_{j-1}}, & \mathrm{if}\;x\geq c_{j-1}\;\mathrm{and}\;x\leq c_{j},\\ 1, & \mathrm{if}\;x>c_{j}\;\mathrm{and}\;x\leq b,\\ 0, & \mathrm{otherwise}. \end{array}\right.$$
An example of a triangular membership function for the variable *V*9 from the Sonar dataset is shown in Figure 1. Based on the corresponding linguistic variables and their values (see Table 1).

### 2.4. Feature Selection Algorithm

The obtained membership value for each created linguistic value constitutes a set, which is subjected to the evaluation and selection of significant value intervals (*FeatureSelectionAlgorithm*). For this purpose, a method based on the random forest paradigm [22] was used. This method is a wrapper method and is a ranking method, i.e., in the process of assessing the significance of features, a ranking of features is created based on a measure of importance. This measure is calculated based on the created set of decision trees. Each tree in such a set is created based on a random sample of data from the original set. In this way, the correlation between dependent variables is minimized. In addition, divisions within the tree are also created based on random subsets of attributes. The tree structure created makes it possible to estimate the importance of an attribute based on decreasing measures of accuracy when an attribute is removed from a node. Since attributes for nodes are selected according to a criterion (in this case, the impurity of the Gini coefficient), we can estimate how each attribute reduces the impurity of a given distribution. The attribute with the largest decrease is placed in the node under consideration. Using this relationship, we can assess the impact of each attribute on the quality of the distributions in the set of trees and, thus, its significance.

Such a methodology is used in the Boruta package [23,24], which allows identifying all relevant attributes from a dataset. It works on an extended dataset containing attributes with random values that have no correlation with the dependent variable. The maximum *MSA* score among the random attributes (shadow attributes) is then determined, which is taken as a threshold for evaluating the importance of features. Attributes whose importances are significantly higher than the *MSA* are placed in the essential attribute group (confirmed features), while those whose importances are significantly lower than the *MSA* are placed in the irrelevant attribute group (rejected features), see Figure 2. This procedure is repeated until all attributes achieve the estimated importance or the algorithm reaches a set limit of random forest runs. Information on the relevance of linguistic variables (*LVs*) allows narrowing down and indicating the relevant subspace of feature values, see Figure 3.

The information about the relevance or irrelevance of the ranges is the basis for creating a binary version of the decision table (see Figure 4). In this table, individual values of the original data are checked for their presence in the relevant (green, value 1) or irrelevant (red, value 0) range. For example, the value of the *V*9 attribute for object number 4 is 0.0598 in the original Sonar dataset. This value is within the range of the created linguistic variable *V*9.LV3, whose minimum value is 0.05525 and maximum value is 0.079025 (see Table 1). This variable was confirmed as an important feature in the feature selection step (see Figure 2), so value 1 will appear in the binary table (see Figure 4). On the other hand, the value of the *V*9 attribute for object number 5 is 0.3564. This value falls within the range of the linguistic variable *V*9.LV20, whose minimum value is 0.299525 and maximum value is 0.520375, and within the range of the linguistic variable *V*9.LV21 whose minimum value is 0.33355 and maximum value is 0.6828 (see Table 1). The variables *V*9.LV20 and *V*9.LV21 were confirmed as irrelevant variables in the feature selection step (see Figure 2), so a value of 0 will appear in the binary table.

## 3. Results

In accordance with the earlier description of the various algorithms and procedures, computational experiments were planned and carried out using the described Sonar dataset and four other well-known datasets.

The detailed results of the analysis of the Sonar dataset are used to present the idea of the operation of the developed algorithms. Table 2 and Figure 5 contain the aggregate results of the *fuzzy feature selection* algorithm, while Table 3 and Figure 6 contain the aggregate results of the *fuzzy object selection* algorithm. In both experiments, the leave-one-out cross-validation approach was applied in the context of splitting into learning and test sets. Decision tree models and Quinlan’s C5.0 algorithm [25] were used to evaluate the quality of classification. Accuracy (*ACC*), sensitivity (true positive rate, *TPR*), specificity (true negative rate, *TNR*), precision (positive predictive value, *PPV*), the Matthews correlation coefficient (*MCC*), and the F1 score (*F1*) were used as measures of classification quality. The calculation formulas for each parameter are presented as follows: (7)$$Accuracy=\frac{TP+TN}{TP+TN+FP+FN}$$
(8)$$Sensitivity=\frac{TP}{TP+FN}$$
(9)$$Specificity=\frac{TN}{FP+TN}$$
(10)$$Precision=\frac{TP}{TP+FP}$$
(11)$$MCC=\frac{TP\;*\;TN-FP\;*\;FN}{\sqrt{\left(TP+FP)(TP+FN)(TN+FP)(TN+FN\right)}}$$
(12)$$F1=\frac{2\;*\;TP}{2\;*\;TP+FP+FN}$$
where: *TP* is the number of results that correctly indicate the presence of a condition or characteristic, *TN* is the number of results that correctly indicate the absence of a condition or characteristic, *FP* is the number of results that wrongly indicate that a particular condition or attribute is present, and *FN* is the number of results that wrongly indicate that a particular condition or attribute is absent.

The analysis of the obtained results of the *FFS* algorithm identifies a full subset of all relevant attributes, which includes 24 features (Table 2). Such a subset allows us to obtain an *ACC* of 0.75 and other parameters at a better level than the original set of 60 features (Figure 5), which allows us to obtain an *ACC* of 0.64. With an increase in the *EPS* coefficient and, consequently, the *threshold*, we observe an improvement in the classification evaluation parameters up to a subset of 16 features, which appears to be the optimal subset. From Table 2 and Figure 5, we can see that equally good results can be obtained using a subset of 9, 12, and 19 relevant features.

On the other hand, the analysis of the obtained results of the *FOS* algorithm allows us to identify a full subset of all relevant features, which includes 204 objects (Table 3). Such a subset allows us to obtain an *ACC* of 0.7 and other parameters at a level similar to that of the original set of 208 objects (Figure 6), which allows us to obtain an *ACC* of 0.73. With the increase in the *EPS* coefficient and, consequently, the *threshold*, we observe an improvement in the classification evaluation parameters, up to a subset of 145 learning objects, which appears to be the optimal subset. Moreover, from both Table 3 and Figure 6, it can be seen that equally good results can be obtained using a subset of 109 relevant objects.

The experimental results presented here allow us to identify a subset of relevant descriptive attributes and a subset of relevant objects in the dataset. This raises the idea of combining these results to identify sub-tables with dimensions suggested by the selected subsets. So, based on the results, four suggested dimensions were selected and classification was performed by determining similar parameters (see Table 4). The results obtained clearly show that the most optimal combination is 187 objects and 19 describing attributes. For this combination, almost all parameters were better than the original dataset. Classification accuracy increased to a value of 0.83 from a value of 0.73 for the original dataset.

Additional analysis of the classification quality was also carried out using subsets of relevant features indicated by standard measures of feature rankings: *Information gain*, i.e., the expected amount of information (reduction of entropy), *gain ratio*, a ratio of the information gain, and the attribute’s intrinsic information, which reduces the bias toward multi-valued features that occur in information gain, the *Gini index*, which is the inequality among values of a frequency distribution, and the *fast correlation-based filter* (FCBF), which is the entropy-based measure, which also identifies redundancy due to pairwise correlations between features. The results of the evaluation of the features of the considered set are shown in Figure 7. The figure includes a ranking of the relevant individual features, considering the mentioned measures. Feature items in the figure are sorted by the information gain ranking values. Each feature, *V*1 to *V*60, has its own ranking. In addition, the last column contains 24 features, which are all relevant, where the *FFS* algorithm is indicated. It can be seen that 16 of the 24 features are also indicated at the top of the ranking (gray highlighting), while the remaining 8 are in lower positions. In addition, classification with the same model using a subset of the 24 highest-ranked features from the ranking yields an accuracy of 0.76, which is slightly better than the results obtained for the 24 features indicated by the *FFS* algorithm (see Table 2), where accuracy is 0.75. A similar subset of features can be selected using gain ratio or Gini index measures (see Figure 7 ). In contrast, the FCBF method identifies only five relevant features: *V*12, *V*9, *V*49, *V*5, and *V*27. Restricting the dataset to these features yields an accuracy of 0.73, which is lower than the other subsets and significantly lower than the identified subset of 19 features and 187 objects (Table 4) for which the best performance in this accuracy of 0.83 is obtained.

### Related Results

A comparison of the results obtained with other studies is possible within a certain range. In the studies, the authors use various evaluation parameters, like *ROC AUC* or the *balanced classification rate*, so we limit the comparison to the *classification accuracy* (*ACC*) parameter. The authors also use different algorithms for learning models (logistic regression (LR), decision tree (DT), random forest (RF), support vector machine (SVM), etc.), and different implementations of them, which may cause difficulties in comparing the results. The results presented meet similar assumptions of the evaluation and model construction. Table 5 shows the compiled results of relevant feature selection and classification accuracy (*ACC*) obtained for five datasets known in the field of machine learning: *Sonar*, *Pima Indians diabetes database* (Pima), *breast cancer Wisconsin diagnostic* (BCWD), *climate model simulation crashes* (Climate), and *single proton emission computed tomography* (SPECTF). These datasets contain only numerical features and have been studied in many publications. The table includes the results obtained for the original set (*ORIG*), for the fuzzy feature selection (*FFS*) set, and the results obtained in other studies using other approaches (*Other*). The results show that the proposed *fuzzy feature selection* algorithm achieves comparable results with other methods. For example, as in Table 5, for the *BCWD* dataset, the original dataset contains 30 descriptive features to obtain a classification accuracy of 0.95. Using the *FFS* algorithm, the number of features can be reduced to 16, with an accuracy of 0.96. Alickovic and Subasi [26], using feature selection based on genetic algorithms [27], obtained 14 relevant features and a classification quality of 0.94. Lopez et al. [28] proposed a framework of the ensemble feature selection (F-EFS), which identified 10 relevant features and a classification quality of 0.92. Neumann et al. [29] proposed their own method of the ensemble feature selection, which integrates eight different feature selection methods and normalizes all individual outputs to a common scale, with an interval from 0 to 1. For the *BCWD* dataset, they obtained 10 relevant features and a classification accuracy of 0.99 using a logistic regression (LR) model. A similar discussion of the results can be made for the other investigated datasets.

Detailed results of the analysis of the four mentioned datasets are presented in Figure 8, Figure 9, Figure 10, Figure 11, Figure 12, Figure 13, Figure 14 and Figure 15. The graphs contain the results of the *ACC*, *TPR*, *TNR*, and *PPV* classification parameters for a given set, combined with the number of features in a given set at a specific value of the *EPS* parameter.

## 4. Discussion

The research presented here yielded preliminary results of a new approach to the problem of selecting relevant attributes and appropriate ranges of their values. In addition, a method for evaluating and selecting a subset of significant objects from the dataset was proposed. These methods are based on evaluating the relevance of ranges of attribute values by applying fuzzy logic. Detailed results obtained on the Sonar dataset show the positive effects of this approach. Out of 208 objects, the algorithm identifies subsets of about 145 and 187 relevant cases, and out of 60 features, it identifies about 9, 12, or 19 relevant features, significantly reducing the dimensionality of the problem and simplifying measurements. As shown in the Results section, the combination of 187 objects and 19 selected important features yields the highest classification quality parameters. All parameters (Table 4) reach a value of about 10% higher than for the full *Sonar* set.

The classification results obtained for the other datasets used (Table 5), resulting from the computational experiments shown in Figure 5, Figure 6, Figure 7, Figure 8, Figure 9, Figure 10, Figure 11, Figure 12, Figure 13, Figure 14 and Figure 15, allow us to assess the effectiveness of the proposed approach at a level equal to or better than the methods proposed in the literature.

By taking into account the discretization operation and feature value fuzzification, the proposed method seems to be more precise in assessing the significance of features. These operations make it possible to assess the relevance of individual value ranges due to linguistic variables, separately. The relevance is a measure of assessing the relevance of the entire attribute. The effectiveness of both the feature and object selection was confirmed by a 10% increase in classification quality parameters. This method has also shown the phenomenon of the more precise identification of relevant features in previous studies. Reference [18] showed that *6* features from the *breast cancer Wisconsin diagnostic* dataset, selected by the selection algorithm in the traditional way, were rejected after analysis with the proposed fuzzy approach, as none of the ranges of the values proved to be significant. It may suggest the effectiveness of the proposed method in the context of identifying a subset of truly relevant attributes among those identified by traditional feature selection methods. The novelty of the proposed approach involves the simultaneous selection of features and objects and horizontal and vertical data-dimensionality reduction.

This approach may find application in the analysis of datasets, where there is a need to identify specific ranges of continuous attribute values. Such datasets exist in the area of medical data, where only selected, narrow ranges of values of diagnostic test results have a significant impact on determining a disease diagnosis. Another example of such an area may be the field of spectrometry [33], where only certain ranges of wavelengths have significant relationships with the dependent variable. In general, such FTIR and Raman spectrometry experiments make it possible to check the absorption of wavelengths of different wavelengths penetrating the biological–chemical sample and tissue under study. The absorption of specific wavelengths can distinguish between different samples, for example, and identify diseases [34]. Application areas can be found in abundance, especially where the data are continuous in nature. The presented method works on such data.

## Figures and Tables

**Figure 1 entropy-25-01223-f001:**
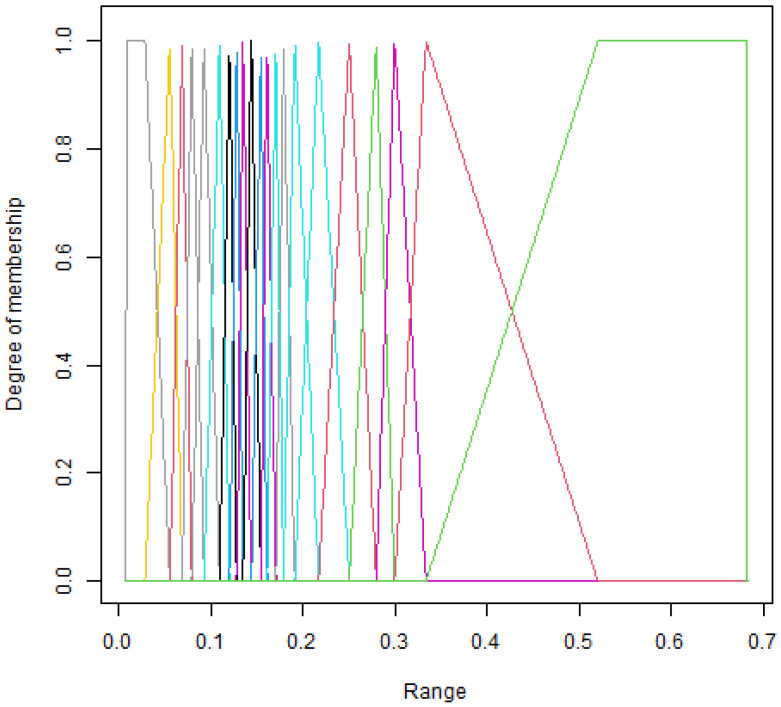
An example of defining a triangular membership function of the value of variable *V*9 from the Sonar dataset based on the designated discretization intervals. The colors correspond to the different linguistic variables (LV) of the *V*9 attribute (see Table 1)

**Figure 2 entropy-25-01223-f002:**
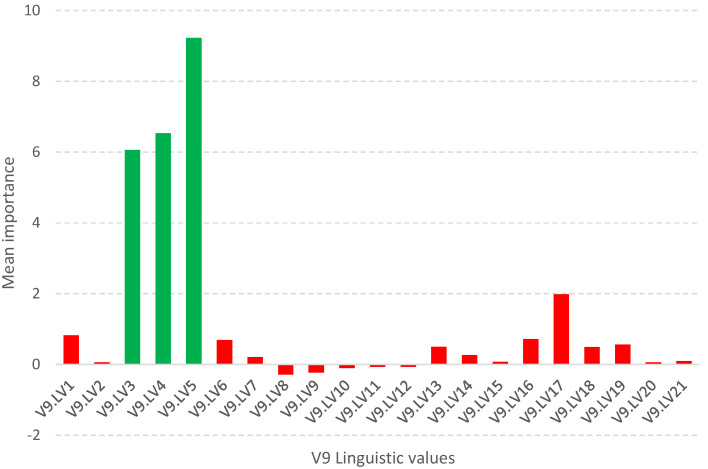
Graph of the average importance value of linguistic variables for the *V*9 attribute from the Sonar dataset. Green variables are those that are confirmed relevant, while red variables are those that are rejected.

**Figure 3 entropy-25-01223-f003:**
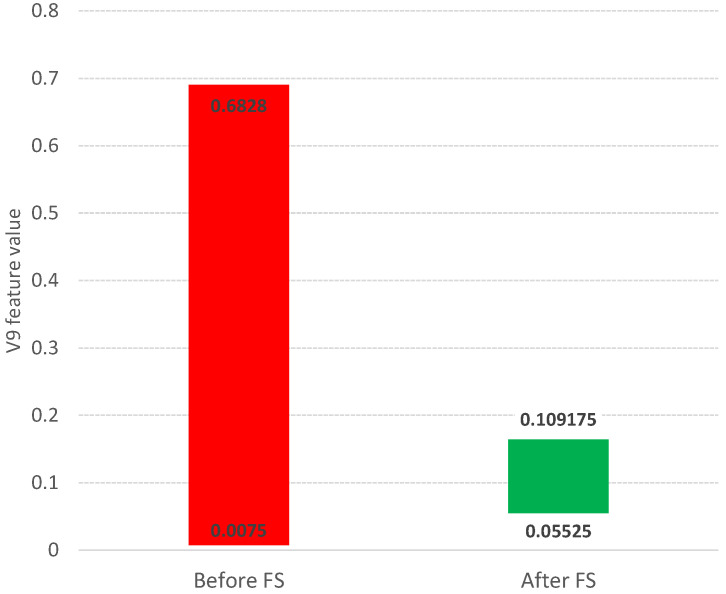
The range of values of the *V*9 original variable (red color) and the values after the selection include linguistic variables (green color).

**Figure 4 entropy-25-01223-f004:**
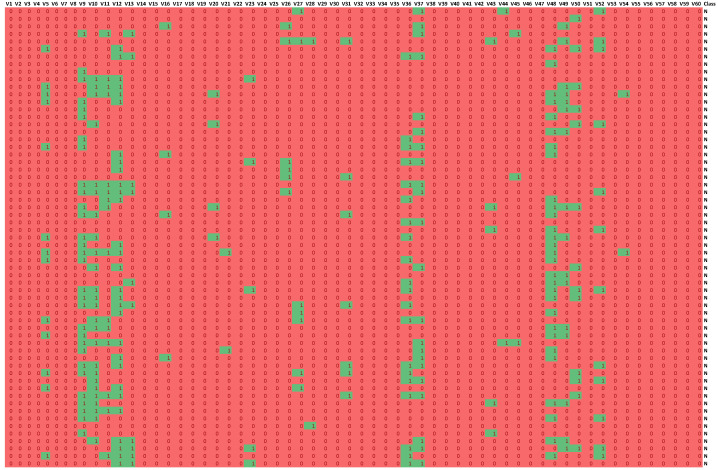
Part of the binary table obtained for the Sonar dataset. The green fields (value 1) indicate the value of an attribute that is in the relevant value range, the red fields (value 0), on the other hand, are the values that are considered irrelevant.

**Figure 5 entropy-25-01223-f005:**
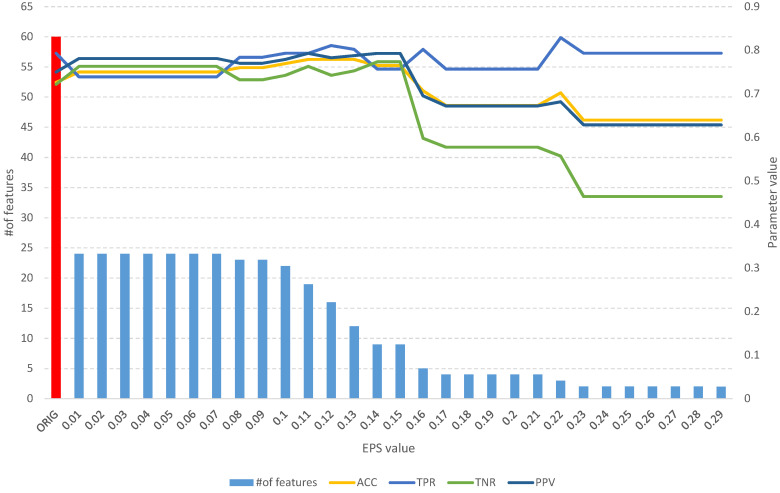
The results of the *fuzzy feature selection* algorithm obtained using the Sonar dataset for different values of the *EPS* parameter, along with parameters for assessing the quality of the classification of the model built on the subset. The red color indicates the results of the original set.

**Figure 6 entropy-25-01223-f006:**
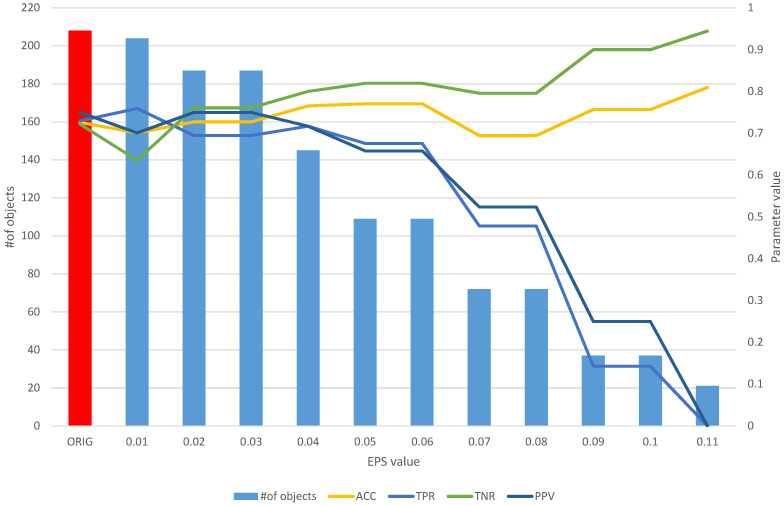
The results of the *fuzzy object selection* algorithm obtained using the Sonar dataset for different values of the *EPS* parameter, along with the parameters for assessing the classification quality of the model built on the subset. The results of the original set are marked in red.

**Figure 7 entropy-25-01223-f007:**
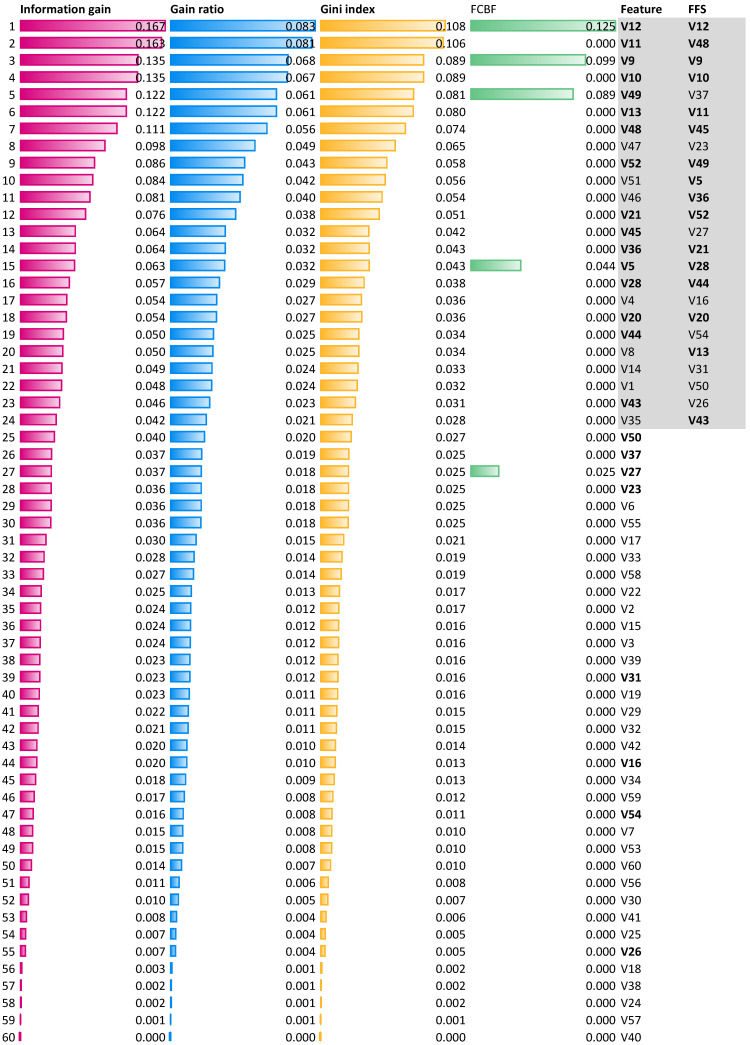
Feature ranking obtained using four parameters, *information gain*, *gain ratio*, *Gini* index, and the *fast correlation-based filter*. The *FFS* column contains all relevant features in the subset indicated by the *fuzzy feature selection* algorithm.

**Figure 8 entropy-25-01223-f008:**
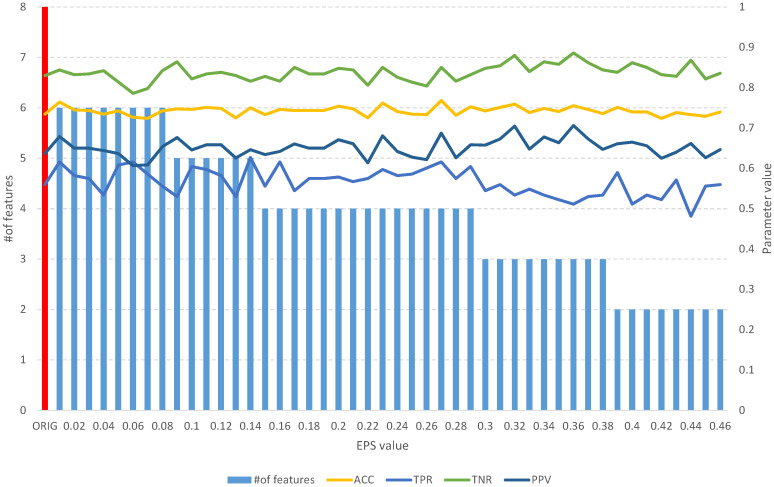
The results of the *fuzzy feature selection* algorithm obtained using the Pima Indians diabetes dataset for different values of the *EPS* parameter, along with parameters for assessing the quality of the classification of the model built on the subset. The red color indicates the results for the original set.

**Figure 9 entropy-25-01223-f009:**
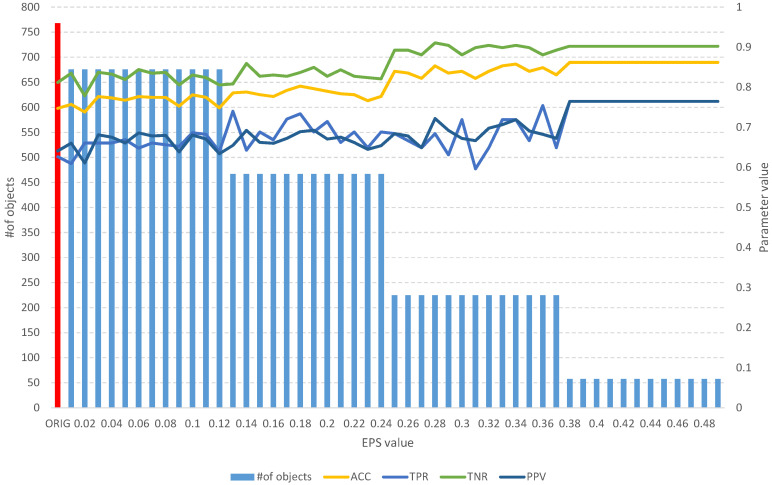
The results of the *fuzzy object selection* algorithm obtained using the Pima Indians diabetes dataset for different values of the *EPS* parameter, along with parameters for assessing the quality of the classification of the model built on the subset. The red color indicates the results for the original set.

**Figure 10 entropy-25-01223-f010:**
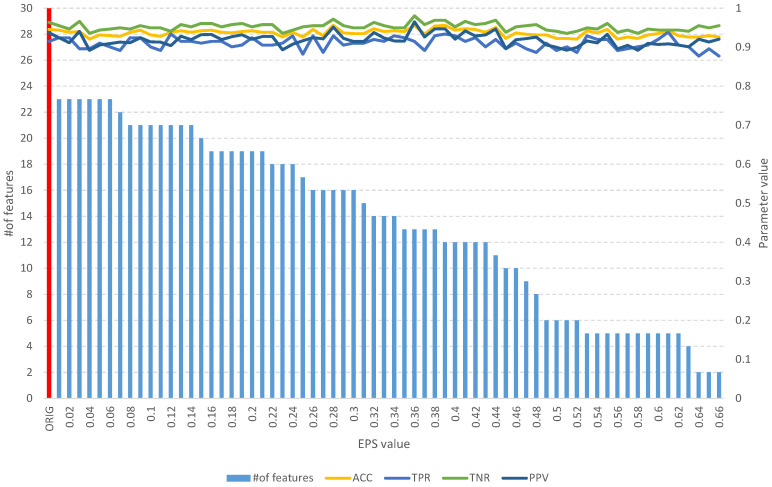
The results of the *fuzzy feature selection* algorithm obtained using the breast cancer Wisconsin diagnostic dataset for different values of the *EPS* parameter, along with parameters for assessing the quality of the classification of the model built on the subset. The red color indicates the results for the original set.

**Figure 11 entropy-25-01223-f011:**
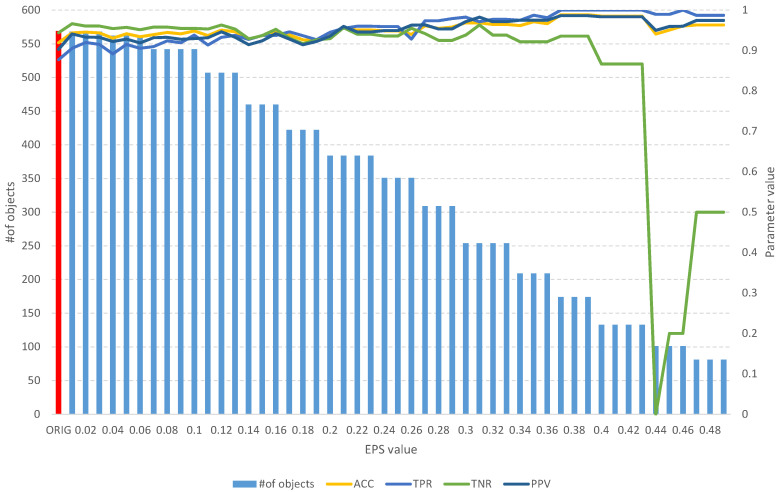
The results of the *fuzzy object selection* algorithm obtained using the breast cancer Wisconsin diagnostic dataset for different values of the *EPS* parameter, along with parameters for assessing the quality of the classification of the model built on the subset. The red color indicates the results for the original set.

**Figure 12 entropy-25-01223-f012:**
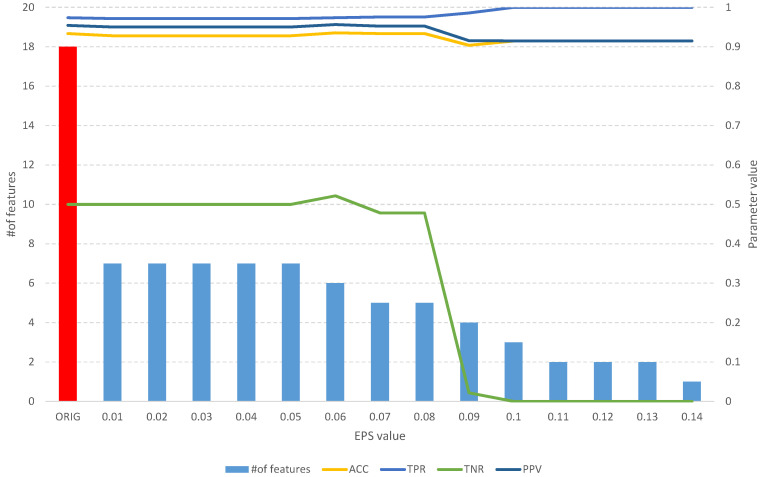
The results of the *fuzzy feature selection* algorithm obtained using the climate model simulation crashes dataset for different values of the *EPS* parameter, along with parameters for assessing the quality of the classification of the model built on the subset. The red color indicates the results for the original set.

**Figure 13 entropy-25-01223-f013:**
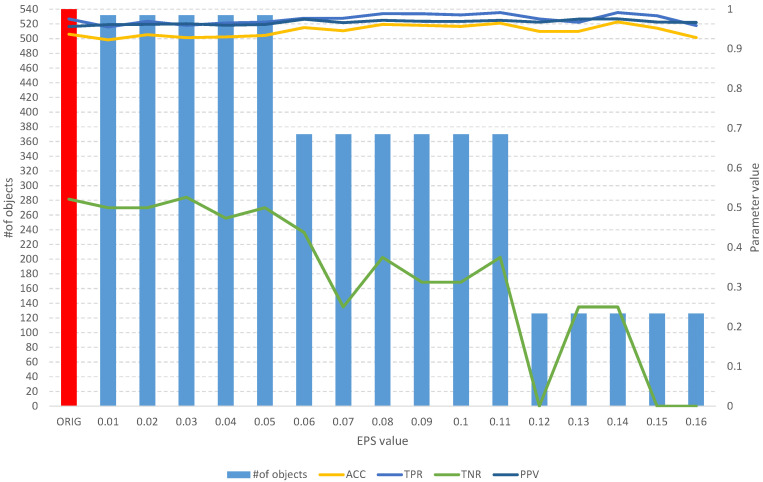
The results of the *fuzzy object selection* algorithm obtained using the climate model simulation crashes dataset for different values of the *EPS* parameter, along with parameters for assessing the quality of the classification of the model built on the subset. The red color indicates the results for the original set.

**Figure 14 entropy-25-01223-f014:**
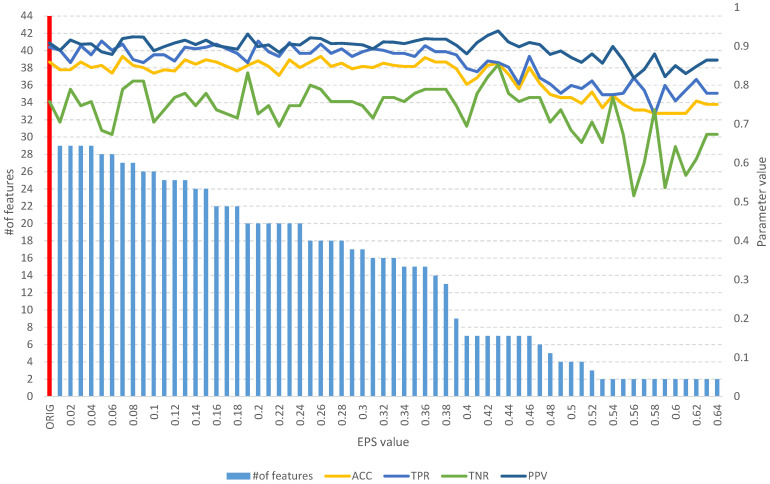
The results of the *fuzzy feature selection* algorithm obtained using the single proton emission computed tomography (SPECTF) dataset for different values of the *EPS* parameter, along with parameters for assessing the quality of the classification of the model built on the subset. The red color indicates the results for the original set.

**Figure 15 entropy-25-01223-f015:**
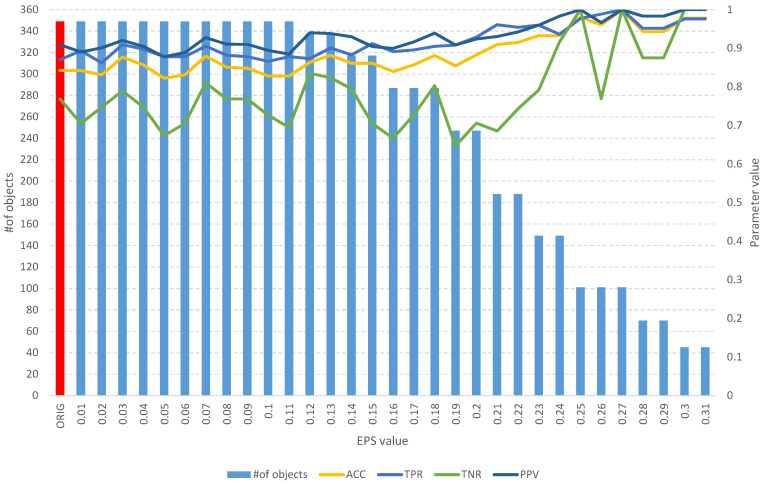
The results of the *fuzzy object selection* algorithm obtained using the single proton emission computed tomography (SPECTF) dataset for different values of the *EPS* parameter, along with parameters for assessing the quality of the classification of the model built on the subset. The red color indicates the results for the original set.

**Table 1 entropy-25-01223-t001:** The discretization intervals of the *V*9 variable, the corresponding linguistic variables, and their minimum and maximum values.

V9 Intervals	Linguistic Values		
0.0075	**name**	**min**	**max**
0.0488	*V*9.LV1	0.0075	0.05525
0.0617	*V*9.LV2	0.02815	0.068525
0.07535	*V*9.LV3	0.05525	0.079025
0.0827	*V*9.LV4	0.068525	0.092325
0.10195	*V*9.LV5	0.079025	0.109175
0.1164	*V*9.LV6	0.092325	0.1202
0.124	*V*9.LV7	0.109175	0.127875
0.13175	*V*9.LV8	0.1202	0.134775
0.1378	*V*9.LV9	0.127875	0.1436
0.1494	*V*9.LV10	0.134775	0.1541
0.1588	*V*9.LV11	0.1436	0.16075
0.1627	*V*9.LV12	0.1541	0.17025
0.1778	*V*9.LV13	0.16075	0.17865
0.1795	*V*9.LV14	0.17025	0.1908
0.2021	*V*9.LV15	0.17865	0.2168
0.2315	*V*9.LV16	0.1908	0.2502
0.2689	*V*9.LV17	0.2168	0.2794
0.2899	*V*9.LV18	0.2502	0.299525
0.30915	*V*9.LV19	0.2794	0.33355
0.35795	*V*9.LV20	0.299525	0.520375
0.6828	*V*9.LV21	0.33355	0.6828

**Table 2 entropy-25-01223-t002:** The results of the *fuzzy feature selection* algorithm obtained using the Sonar dataset for different values of the *EPS* parameter, along with parameters for assessing the quality of the classification of the model built on the subset. The best results are in bold.

No. of Features	EPS	thresholdFFS	ACC	TPR	TNR	PPV	MCC	F1
24	0.01	2.08	0.75	0.74	0.76	0.78	0.5	0.76
24	0.02	4.16	0.75	0.74	0.76	0.78	0.5	0.76
24	0.03	6.24	0.75	0.74	0.76	0.78	0.5	0.76
24	0.04	8.32	0.75	0.74	0.76	0.78	0.5	0.76
24	0.05	10.40	0.75	0.74	0.76	0.78	0.5	0.76
24	0.06	12.48	0.75	0.74	0.76	0.78	0.5	0.76
24	0.07	14.56	0.75	0.74	0.76	0.78	0.5	0.76
23	0.08	16.64	0.76	0.78	0.73	0.77	0.52	0.78
23	0.09	18.72	0.76	0.78	0.73	0.77	0.52	0.78
22	0.1	20.80	0.77	0.79	0.74	0.78	0.54	0.79
19	0.11	22.88	**0.78**	0.79	0.76	0.79	**0.56**	0.79
16	0.12	24.96	**0.78**	0.81	0.74	0.78	0.55	**0.8**
12	0.13	27.04	**0.78**	0.8	0.75	**0.79**	**0.56**	0.79
9	0.14	29.12	0.76	0.76	**0.77**	**0.79**	0.53	0.77
9	0.15	31.20	0.76	0.76	**0.77**	**0.79**	0.53	0.77
5	0.16	33.28	0.71	0.8	0.6	0.7	0.41	0.74
4	0.17	35.36	0.67	0.76	0.58	0.67	0.34	0.71
4	0.18	37.44	0.67	0.76	0.58	0.67	0.34	0.71
4	0.19	39.52	0.67	0.76	0.58	0.67	0.34	0.71
4	0.2	41.60	0.67	0.76	0.58	0.67	0.34	0.71
4	0.21	43.68	0.67	0.76	0.58	0.67	0.34	0.71
3	0.22	45.76	0.7	**0.83**	0.56	0.68	0.4	0.75
2	0.23	47.84	0.64	0.79	0.46	0.63	0.27	0.7
2	0.24	49.92	0.64	0.79	0.46	0.63	0.27	0.7
2	0.25	52.00	0.64	0.79	0.46	0.63	0.27	0.7
2	0.26	54.08	0.64	0.79	0.46	0.63	0.27	0.7
2	0.27	56.16	0.64	0.79	0.46	0.63	0.27	0.7
2	0.28	58.24	0.64	0.79	0.46	0.63	0.27	0.7
2	0.29	60.32	0.64	0.79	0.46	0.63	0.27	0.7
original set	-	-	0.64	0.79	0.46	0.63	0.27	0.7

**Table 3 entropy-25-01223-t003:** The results of the *fuzzy object selection* algorithm obtained using the Sonar dataset for different values of the *EPS* parameter, along with parameters for assessing the quality of the classification of the model built on the subset. The best results are in bold.

No. of Objects	EPS	thresholdFOS	ACC	TPR	TNR	PPV	MCC	F1
204	0.01	0.6	0.7	**0.76**	0.64	0.7	0.4	**0.73**
187	0.02	1.2	0.73	0.69	0.76	**0.75**	0.46	0.72
187	0.03	1.8	0.73	0.69	0.76	**0.75**	0.46	0.72
145	0.04	2.4	0.77	0.72	0.8	0.72	**0.52**	0.72
109	0.05	3	0.77	0.68	0.82	0.66	0.49	0.67
109	0.06	3.6	0.77	0.68	0.82	0.66	0.49	0.67
72	0.07	4.2	0.69	0.48	0.8	0.52	0.28	0.5
72	0.08	4.8	0.69	0.48	0.8	0.52	0.28	0.5
37	0.09	5.4	0.76	0.14	0.9	0.25	0.05	0.18
37	0.1	6	0.76	0.14	0.9	0.25	0.05	0.18
21	0.11	6.6	**0.81**	0	**0.94**	0	-0.09	-
original set	-	-	0.73	0.73	0.72	0.75	0.45	0.74

**Table 4 entropy-25-01223-t004:** Classification results obtained for the selected combinations of the number of relevant objects and features. The best results are in bold.

No. of Objects	No. of Features	ACC	TPR	TNR	PPV	MCC	F1
187	9	0.8	0.84	0.77	0.78	0.61	0.81
**187**	**19**	**0.83**	**0.87**	**0.8**	0.81	**0.67**	**0.84**
145	9	0.81	0.86	0.73	0.82	0.6	**0.84**
145	19	0.81	0.85	0.75	**0.83**	0.6	**0.84**
original set	60	0.73	0.73	0.72	0.75	0.45	0.74

**Table 5 entropy-25-01223-t005:** Classification accuracy obtained for the selected number of relevant features using five different datasets.

Dataset	Parameters	ORIG	FFS	Other
Sonar	No. of features	60	12	10 [28], 24 [29], 49 [30]
(60 features, 208 objects)	ACC	0.73	0.78	0.74 (DT) [28], 0.86 (LR) [29], 0.77 (RF) [30]
Pima	No. of features	8	4	3 [31], 8 [30]
(8 features, 768 objects)	ACC	0.73	0.77	0.73 (DT) [31], 0.72 (RF) [30]
BCWD	No. of features	30	16	14 [26], 10 [28], 10 [29]
(30 features, 569 objects)	ACC	0.95	0.96	0.94 (DT) [26], 0.92 (DT) [28], 0.99 (LR) [29]
Climate	No. of features	18	5	8 [32], 7 [30]
(18 features, 540 objects)	ACC	0.93	0.93	0.97 (SVM) [32], 0.92 (RF) [30]
SPECTF	No. of features	44	18	10 [28], 19 [29]
(44 features, 349 objects)	ACC	0.86	0.87	0.75 (DT) [28], 0.86 (LR) [29]

## Data Availability

Data are available in a publicly accessible repository that does not issue DOIs. Publicly available datasets were analyzed in this study. These data can be found here: [https://data.world/uci/connectionist-bench-sonar-mines-vs-rocks] accessed on 1 June 2023.

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
