# Peer review of "Application of the Fuzzy Approach for Evaluating and Selecting Relevant Objects, Features, and Their Ranges"

_entropy, 2023, doi:10.3390/e25081223_

Round 1
Reviewer 1 Report
This manuscript focuses on improving the fundamental AI algorithms, which fall outside the scope of this journal. The term 'entropy' does not appear in the content. it is necessary to include some considerations about entropy in relation to the algorithm.
From 78-83, there is no explanation of the fuzzification method. Does A_Fuzz imply the function converting the data to discrete values, and assigned to linguistic variables, and then converting them to binary cords? Why the introduction of this function is optimised for adequate fuzzification? This reviewer cannot follow the presented algorithm’s features. For this, The author should list the reference of the basic idea of this fuzzification.
Finally, why is it applicable to disease diagnosis and spectroscopy? Need detailed explanation.
The discussion is also very short, and there is no mention of the novelty of this research or the limitations of the research.
This manuscript seems to be a draft. The conclusion section remains in submission format. Therefore, this manuscript does not contain any conclusion.
"5. Conclusions
This section is not mandatory, but can be added to the manuscript if the discussion is unusually long or complex."
Author Response
I kindly thank you for your time and substantive comments on my manuscript. I have included responses to each comment below and a revised version of the manuscript is attached.
Point 1: This manuscript focuses on improving the fundamental AI algorithms, which fall outside the scope of this journal. The term 'entropy' does not appear in the content. it is necessary to include some considerations about entropy in relation to the algorithm.
Response 1: Additional studies have been added to the manuscript comparing the selection results of the proposed method with existing standard methods using entropy in the feature ranking and selection process like Information Gain, Gain Ratio, Gini index and Fast Correlation Based Filter (Figure 7). I hope, this will be a sufficient argument for fulfilling the scope of journal fit in information theory. Please, see the attachment.
Point 2: From 78-83, there is no explanation of the fuzzification method. Does A_Fuzz imply the function converting the data to discrete values, and assigned to linguistic variables, and then converting them to binary cords? Why the introduction of this function is optimised for adequate fuzzification? This reviewer cannot follow the presented algorithm’s features. For this, The author should list the reference of the basic idea of this fuzzification.
Response 2: Lines 78-83 contain only a rough description of the algorithm whose pseudocode is also included in the manuscript. A detailed description of its various stages is provided in the following sections 2.2, 2.3, and 2.4. The details of the fuzzification process are just presented in section 2.3. This process allowed us to obtain information about the relevance of the various ranges of attribute values.
Point 3: Finally, why is it applicable to disease diagnosis and spectroscopy? Need detailed explanation.
Response 3: It is listed as an example of application areas. This approach may find application in analysis of datasets where there is a need to identify specific ranges of continuous attribute values. Such datasets exist in the area of medical data, where only selected, narrow ranges of values of diagnostic test results have a significant impact on determining a disease diagnosis. Another example of such area may be the field of spectrometry, in which only certain ranges of wavelengths have a significant relationship with the dependent variable. In general, such FTIR and Raman spectrometry experiments make it possible to check the absorption of wavelengths of different wavelengths penetrating the biological-chemical sample, tissue under study. Absorption of specific wavelengths can distinguish between different samples, for example, identifying a disease. Application areas can be found in abundance, especially where the data is continuous in nature. The presented method works on just such data.
Point 4: The discussion is also very short, and there is no mention of the novelty of this research or the limitations of the research.
Response 4: The Discussion section was expanded to include additional elements in response to reviewers' suggestions. Please, see the attachment.
Point 5: This manuscript seems to be a draft. The conclusion section remains in submission format. Therefore, this manuscript does not contain any conclusion.
"5. Conclusions
This section is not mandatory, but can be added to the manuscript if the discussion is unusually long or complex."
Response 5: I kindly apologize for this error. It is the result of not using the comment function in the article template. It has been removed.

Reviewer 2 Report
The paper’s subject is relevant and interesting. The selection of attributes in data mining problems, particularly classification, is essential and influences classification efficiency. The author proposes two new algorithms for selecting relevant features and evaluating and selecting a subset of relevant objects in a dataset. The application of fuzzy logic allows the achievement of acceptable results compared to the known approaches for attribute selection. The experimental study is presented and confirms the efficiency of the proposed algorithms.
In my opinion, the paper has limited analysis of the problem state. I’d like to recommend extending the problem state's analysis, considering new approaches in the attribute selection, and adding more new studies of last years in this domain to the list of references. In particular, the studies can be considered :
Zaitseva, E., Rabcan, J., Levashenko, V., Kvassay, M., Importance analysis of decision making factors based on fuzzy decision trees,
Applied Soft Computing, 2023, 134, 109988
Li, Q., Rong, Y., Pei, Z., Ren, F., A novel linguistic decision making approach based on attribute correlation and EDAS method, Soft Computing, 2023, 27(12), pp. 7751-7771
Author Response
Dear rewiewer,
I kindly thank you for your time and valuable feedback on my manuscript.
In the revised version of the manuscript, I have tried to outline the problem more broadly and have included suggested literature items. Please see the attachment.

Reviewer 3 Report
The threshold was not clearly defined. Also, how were you getting to the threshold -- what was the criteria for the threshold chosen?
More needs to be added to the conclusion.
Section 2.1 A reference is missing after Sonar in the second sentence.
Please read the paper carefully -- in at least a couple places there were periods where there should be commas.
Author Response
Dear reviewer
I kindly thank you for your time and valuable comments on my manuscript.
In order to explain in a little more detail how we determine the threshold value, I have added an additional description in Section 2 lines 130-155. ThresholdFFS is a parameter that defines, so to speak, the range of significant ranges of the trait value. If ThresholdFFS is 20.80, then those features that have more than 20.80 important value ranges are considered relevant. In our case, 22 features meet this condition. The analogy is with ThresholdFOS for the selection of relevant features. If, for example, ThresholdFOS is 3.60 then those that have more than 3.60 important value ranges are recognized as relevant objects. In our case, 109 objects meet this condition.
The Discussion section has been further developed.
The missing reference has been added.
Please see the attachment.
